# A Practical Approach to Using Integrated Knowledge Translation to Inform a Community-Based Exercise Study

**DOI:** 10.3390/ijerph17113911

**Published:** 2020-06-01

**Authors:** Kirsten Suderman, Naomi Dolgoy, Janice Yurick, Christopher Sellar, Kathryn Nishimura, S. Nicole Culos-Reed, Anil A. Joy, Margaret L. McNeely

**Affiliations:** 1Department of Physical Therapy, University of Alberta, 2-50 Corbett Hall, Edmonton, AB T6G 2G4, Canada; miazga@ualberta.ca (K.S.); dolgoy@ualberta.ca (N.D.); csellar@ualberta.ca (C.S.); knishimu@ualberta.ca (K.N.); 2Cross Cancer Institute, Alberta Health Services, 11560 University Avenue, Edmonton, AB T6G 1Z2, Canada; frmace@ualberta.ca (J.Y.); Anil.Joy@albertahealthservices.ca (A.A.J.); 3Faculty of Kinesiology, University of Calgary, 2500 University Drive NW, Calgary, AB T2N 1N4, Canada; nculosre@ucalgary.ca

**Keywords:** exercise, physical therapy, cancer, knowledge translation, implementation, barriers

## Abstract

Background: Our aim was to understand cancer survivor needs prior to, and following the Alberta Cancer Exercise (ACE) pilot randomized trial as a means to inform implementation of a province-wide cancer-specific, community-based exercise program. Methods: Questionnaires and semi-structured stakeholder engagement sessions were conducted with cancer survivors to explore preferences, barriers and facilitators/benefits at two timepoints: (1) pre-ACE: prior to initiation of the ACE pilot trial (*n* = 13 survivors and *n* = 5 caregivers); and (2) post-ACE: following participation in the ACE pilot trial (*n* = 20 survivors). Descriptive statistics were used to summarize quantitative data from questionnaires. Stakeholder engagement data were analyzed using a framework analysis approach. Emergent themes were then mapped to actionable outcomes. Results: Pre-ACE, survivors indicated a preference for exercise programs that were (1) supervised by exercise specialists knowledgeable about cancer, (2) included support from other health care providers, (3) were held in community locations that were easily accessible. Post-ACE, participants identified (1) a lack of exercise counseling from health care providers, (2) the need for earlier introduction of exercise in the care pathway, and (3) supported referral to exercise programming. Conclusions: An integrated knowledge translation approach identified actionable outcomes to address survivor needs related to exercise in clinical cancer and community-based contexts.

## 1. Introduction

### 1.1. Cancer and Exercise Evidence

There is a growing population of individuals surviving cancer. In Canada, over the past 20 years, the predicted five-year age-standardized net survival rate has increased by 8%, with 63% of all cancer patients surviving at least five years post-diagnosis [1]. The term cancer survivor encompasses the entire continuum of cancer care, and is defined as any person diagnosed with cancer from the initial point of diagnosis, until death [2]. Advancements in treatments, screening and technologies have led to improved survival rates, and resulted in a growing population of cancer survivors living with cancer-related treatment long-term and late effects, including decreased physical and psychological functioning and overall quality of life [3]. Exercise is an intervention that has shown benefit in addressing the supportive care needs of this growing survivor population, with strong evidence establishing the positive effects of exercise on symptom management, physical and psychosocial well-being and health-related quality of life [4]. Moreover, current evidence suggests a positive association between physical activity and cancer outcomes of recurrence, cancer-specific mortality and all-cause mortality, predominately in prostate, breast and colon cancer [4,5,6].

Despite the known benefits of exercise, the majority of survivors do not meet public health guidelines for physical activity, with survivors engaging in significantly less physical activity compared to those without a cancer diagnosis [7,8]. Moreover, only 10% of cancer survivors were found to be physically active during treatment and only 20%–30% active post-treatment and into survivorship [9]. The aging process and existing comorbid diseases present additional challenges to physical activity, with older cancer survivors less likely to meet physical activity guidelines than their younger counterparts [10].

In more recent years, there has been a global initiative to encourage health care providers (HCPs) to include physical activity counseling and exercise referral into the care plans of patients with chronic disease [11]. To date, however, there is limited evidence supporting implementation of feasible, effective and sustainable exercise counselling and referral practices, and program delivery within the clinical oncology setting. Research into the preferences, barriers and facilitators of exercise from the survivors’ perspective has revealed complex areas of need that are only partly addressed by current exercise guidelines [12,13,14,15].

### 1.2. Research Context of the Clinical Team

The Alberta Cancer Exercise (ACE) trial was a multi-centre randomized controlled pilot trial, performed to examine the feasibility and preliminary effectiveness of a cancer-specific community-based exercise program [16]. The ACE trial (*N* = 80) was conducted from April 2015 to November 2017, and had two sites, Edmonton (*n* = 46) and Calgary (*n* = 34) in Alberta, Canada. The ACE pilot trial included an integrated knowledge translation (iKT) plan and series of iKT substudies aimed to identify the needs of key informants and address specific gaps related to our local provincial context. The ACE iKT strategy involved drawing on the perspectives of urban and rural cancer survivors and caregivers to inform the design of the pilot trial, and feedback from ACE pilot trial participants after study completion. The purpose of this paper is to share the findings related to survivor reported exercise preferences, barriers, and facilitators before and after participation in the ACE pilot trial, and to describe how the findings informed the design of the current five-year ACE Hybrid-Effectiveness Implementation Study [17].

## 2. Materials and Methods

### 2.1. Study Design

This study utilized a multi-method iKT approach to better understand survivors’ preferences, barriers and facilitators to implementation of a province-wide community-based exercise program. The Knowledge-To-Action (KTA) model was used to inform the implementation process for the study [18], and aligned with cycle phases associated with (1) adapting knowledge to the local context and (2) assessing barriers and facilitators to cancer-specific exercise programming in our local context. A multi-method design including both quantitative and qualitative data collection, was used to enable a richer description of possible determinants that may influence successful implementation of the program [19]. The study consisted of pre- and post-ACE questionnaires and stakeholder engagement group (SEG) sessions consisting of cancer survivors and, in the pre-ACE SEGs, caregivers. To achieve data saturation, we aimed for a sample size of 10–15 cancer survivors at each time point, with 5–8 participants per session.

The questionnaires included collection of medical and demographic information, and current exercise behavior and exercise preferences. At both time points, the questionnaires were anonymous to encourage accurate reporting of survivor opinions towards exercise. The post-ACE questionnaire was optional, and included additional questions related to ACE pilot program satisfaction [20]. The questionnaires were used to provide numeric and descriptive data for the purposes of analyses.

SEGs were conducted as a formative evaluation to inform program design and improvement [21], and followed a semi-structured interview guide developed by the clinical research team. The SEGs were used to better understand survivors’ needs related to implementing a cancer-specific community-based exercise program. Pre- and post-SEGs were conducted over multiple groups until data saturation was achieved. Each SEG was led by a trained facilitator, along with experts in cancer exercise physiology and physical therapy. Ethics approval for the study was granted by the Health Research Ethics Board of Alberta: Cancer Committee (HREBA.CC-14-0153) (Figure 1).

### 2.2. Data Collection

Cancer survivors and caregivers were recruited for the pre-ACE SEG using a convenience sample of urban and rural survivors and caregivers from the Cross Cancer Institute, identified by staff from physical therapy and/or radiation therapy. Survivors and caregivers taking part in the pre-ACE SEGs and anonymous questionnaires were not participants in the ACE pilot randomized trial. Post-ACE SEGs involved a convenience sample of ACE pilot trial participants. Upon completion of the ACE pilot trial, participants were invited to participate in SEG sessions and to complete an additional anonymous satisfaction questionnaire. Caregivers were not included in the post-ACE SEGs, due to high ACE pilot trial participant interest in the SEGs. Survivor input was collected at two time points: (1) prior to ACE pilot study initiation, July 2013 (pre-ACE); (2) post-ACE pilot (Edmonton site), May 2016 (post-intervention). Informed consent was obtained from participants at each time point. Each SEG session consisted of 5–8 participants, and lasted approximately 90 min. The SEG sessions involved discussion on the topics of the preferences, barriers and benefits/facilitators to programming, with questions differing slightly at the two time points (see Appendix A). The SEG sessions started with participants providing individual written responses for brainstorming activities, followed by small group discussions. Small group membership was established a priori to ensure representation in each group of males and females, different tumor types and geographical locations, and in the Pre-ACE SEGs, involvement of at least one caregiver in each group. Additional probing questions were supplied when participants perceived questions to be unclear, or further information was needed. Independent observers were used to transcribe the discussion. Given the primary implementation focus, less in-depth analyses were planned, thus, independent observers were used to take abridged transcripts of notes using a laptop during the SEG sessions [21]. The pre-ACE SEG involved three sessions to reach saturation, while the post-ACE SEGs had four sessions until saturation was achieved. Questionnaires were completed at the conclusion of each SEG. The questionnaires for the post-ACE SEG participants were optional, given the high study burden of questionnaires associated with the ACE pilot. 

### 2.3. Data Analysis

Data from the SEGs were analyzed using framework analysis, a form of content analysis for identifying commonalities and differences in qualitative data, with a defining feature being structured matrix outputs involving rows and columns of summarized data [22]. Framework analysis originated in large scale policy research, and has become increasingly popular in multi-disciplinary medical and health applied policy research to meet specific information needs with actionable outcomes [23]. Framework analysis is not aligned with one epistemological or theoretical approach, but can be adapted to various qualitative approaches that aim to generate themes, and offers valuable insight to inform implementation strategies [23].

After each SEG was completed, abridged transcriptions and written responses by participants were collected. Two researchers independently analyzed the abridged transcriptions and written materials from participants pre-ACE (CS, MM), and post-ACE (KS, MM). After initial coding, researchers collaborated to amend and refine codes, and develop mapping framework tables in relation to barriers, facilitators and preferences towards exercise. After the data were coded, codes were mapped to respective frameworks and researchers reviewed the codes to identify prevalent themes. The themes were then reviewed, defined and categorized. This process occurred after each SEG session, until data saturation was achieved. Identified themes from the pre-ACE SEG informed the design of the ACE pilot randomized trial, and themes from both ACE SEG sessions informed the design of the current five-year Alberta Cancer Exercise Hybrid Effectiveness-Implementation Study [16,17].

Data from the pre- and post-questionnaires included both continuous and categorical variables. Basic descriptive statistics including frequencies, percentages and counts were calculated. Linking qualitative and quantitative data analysis involved building on quantitative descriptive statistical patterns through qualitative thematic findings that revealed the perspective and thought processes of participants [23].

## 3. Results

### 3.1. Description of Participants

A total of *N* = 33 distinct survivors and *N* = 5 caregivers took part in the study. Thirteen cancer survivors and five caregivers participated in pre-ACE SEG; and twenty ACE pilot participants engaged in post-intervention SEG groups and eighteen completed the post-ACE questionnaire. Seventeen ACE trial participants completed an anonymous optional post-study satisfaction questionnaire. The majority of participants in attendance at both SEGs were breast cancer survivors, with 54% pre-ACE, and 60% post-ACE. The second most common tumor type at both the pre- and post-ACE SEGs was head and neck cancer (23%, 20%), followed by gastrointestinal (16%, 10%), and lymphoma (8%, 5%). The most commonly reported age of participants identified in the category of 55–69 years of age (39%). The majority of survivors were female, with 69% and 85% pre- and post-ACE, respectively. Both pre-ACE and post-ACE SEGs consisted primarily of survivors who had received combined treatment including surgery, chemotherapy and radiation, 54% and 45%, respectively, with the second most common combined treatments pre-ACE involving surgery and chemotherapy (15%) and surgery radiation (15%), and post-ACE involving combined surgery and radiation (35%) (Table 1).

### 3.2. Pre-ACE Questionnaire Findings

Eighty-five percent of survivors completing the pre-ACE questionnaire indicated the need for exercise counseling in the clinical setting, with 77% reporting that exercise had not been discussed at any point during their cancer treatment or follow-up visits. The preferred location for exercise counseling to take place was at the cancer centre (54%), with delivery by multiple HCPs, including a strong preference for counseling from an exercise professional (62%). All participants preferred face-to-face counseling (77%) or written materials (15%), with delivery of counseling at multiple time points along the cancer treatment trajectory (69%) (Table 2).

### 3.3. Post-ACE Questionnaire Findings

Overall, participant satisfaction with the ACE pilot trial was 91%. From the post-study satisfaction questionnaire, only 7% of participants indicated that their oncologist or HCP had referred them to the ACE trial, with 93% indicating self-referral. Participants reported symptoms as somewhat improved to very much improved regarding physical functioning (88%), muscle strength (82%), overall quality of life (76%), fatigue (65%), energy levels (65%), activities of daily living (59%) and recovery from treatment (53%).

From the post-ACE questionnaire, a majority of participants (56%) preferred the ACE pilot trial format of two exercise sessions per week for a 60 min duration, with 94% preferring moderate-intensity exercise. Eighty-two percent indicated a preference for a combination of supervised and unsupervised exercise, with 77% preferring to exercise with other cancer survivors and 100% preferring a combination of aerobic and resistance exercise. Of note, 83% preferred to continue exercising at the location of the pilot trial and 76% perceived little to no difficulty in continuing to exercise independently post-intervention. The most frequently self-reported barriers to exercise (reported as often to very often) were muscle weakness, reported by 28% of participants, followed by symptoms of fatigue (17%), pain (11%), lack of enjoyment (11%) and weather (11%) (Table 3).

### 3.4. Stakeholder Engagement Group Findings

#### 3.4.1. Exercise Preferences

At both pre-ACE and post-ACE SEGs, participants consistently indicated a preference for supervised and supported exercise programming that was accessible, affordable, and variable. All survivors indicated a preference for exercise programming that (1) was supervised by exercise specialists knowledgeable about cancer, (2) included support from other HCPs (e.g., physical therapy), and (3) had a variety of exercise delivery options, and (4) was held in community-based locations that were easily accessible.

Post-ACE SEG participants indicated a unique preference for tumor-specific programming to better address impairments and for the option to include caregivers in exercise programming. Post-ACE SEG participants also expressed the need for (1) better HCPs awareness and promotion of the exercise programming; (2) exercise counselling to occur earlier in the cancer treatment time period; and (3) formal referral to cancer-specific exercise programming.

Post-ACE SEG participants also identified a new theme of communication. Specifically, participants identified a need for multi-directional communication between survivors, health care professionals (HCPs) and exercise specialists to integrate tailored exercise through cancer treatment and survivorship.

“*I may not have wanted or been able to take part in exercise when I was on treatment, but I would have liked to have known about the program and the option to take part later*”.(ACE trial participant with lymphoma)

#### 3.4.2. Barriers towards Exercise

Time was consistently identified as the main barrier to exercise across SEGs at both time points. Pre-ACE, participants identified risk of injury and lack of familiarity with use of exercise equipment and machines as a barrier to exercise. Unique barriers identified post-ACE included: (1) concerns over potential exposure to bacteria and viruses in a public fitness facility when immunocompromised; (2) return to work scheduling and ongoing medical appointments conflicting with ability to attend exercise sessions. Accessibility issues to the ACE pilot trial included the downtown location of the community fitness centre due to parking fees, traffic, and seasonal winter road conditions.

“*When I was on chemotherapy and my counts were low, I did not attend class*”.(ACE trial participant with breast cancer)

#### 3.4.3. Facilitators and Benefits of Exercise

Participant feedback at both pre- and post-ACE SEG time periods indicated benefits for physical fitness, wellness autonomy (control over one’s health), increased knowledge regarding safe exercise practices, understanding cancer related physical limitations and motivation towards exercise.

“*I knew it would be good for me, but I didn’t think I would look forward to going or enjoy the sessions*”.(ACE trial participant with breast cancer)

Post-ACE SEG participants specifically identified exercise programming that addressed tumor specific physical impairments (breast, head and neck and neurological tumors) as a facilitator towards exercise. Other specific post-ACE exercise facilitators included regular feedback from fitness assessments, and HCP support from an on-site physical therapist. A unique post-ACE benefit was exercise programming that was adapted/modified for cancer related symptoms of lymphedema, chemotherapy induced peripheral neuropathy, and shortness of breath. Additional unique identified benefits post-intervention included: (1) a needed “wake-up call” on low fitness levels; (2) the benefit of support and mentorship of fellow cancer survivor peers; and (3) increased confidence with use of resistance exercises and fitness equipment.

“*The program has given me the confidence to re-enter society*”.(ACE trial participant with head and neck cancer)

Exercise preferences, barriers and facilitators/benefits from both timepoints were synthesized to inform themes. Three emergent themes were identified related to the cancer care setting and four themes related to the community context (Figure 2). Findings were mapped to nine potential actionable items (Table 4).

## 4. Discussion

The purpose of this study was to utilize a multi-method iKT approach to identify cancer survivors’ exercise preferences, barriers and facilitators to inform implementation of a cancer-specific community-based exercise program. The central premise of iKT is that involvement of knowledge users, in this case cancer survivors, throughout the research process leads to research and outputs that are more applicable and helpful to the knowledge users (survivors) [24]. As a formal methodological approach does not yet exist for iKT, we chose a multi-method approach that allowed us to efficiently identify key actionable strategies in a time-sensitive project. By using this multi-methods approach, we were able to provide more depth to the participant perspective, allowing a comprehensive understanding of local exercise preferences, barriers and facilitators [23]. To our knowledge, this is the first study to explore using a multi-method iKT approach to inform future implementation of cancer-specific exercise programming in a community-based setting.

A primary finding of this study was the identified gap regarding the lack of exercise counseling and referral provided to survivors in the cancer clinical setting. A new theme of communication, encompassing survivors, HCPs and exercise specialists, was identified by participants as a prevalent patient need. Specifically, participants identified a preference and need for earlier introduction and integration of exercise counselling into care pathways, including referral to appropriate cancer-specific exercise programming during cancer treatment and into survivorship. The findings from the pre-ACE questionnaire suggest that few survivors received exercise counseling from their HCPs, and the post-ACE questionnaire reflected the gap in terms of exercise referral, with only 7% indicating referral from HCPs. The reported lack of exercise counselling and referral is not unique to our site. Another Canadian study reported that less than 20% of survivors had received education on the importance of exercise from any HCP at any point in the course of their cancer treatment [25]. Further, 83% and 88% of patients reported not receiving any exercise counseling from their oncologist and primary cancer nurse, respectively. Additional studies suggest HCP exercise counselling rates are low, with less than 25% of oncology physicians actually referring survivors to exercise programming [26,27]. These findings are not surprising given the lack of availability of cancer-specific exercise programming.

This study brings forward a unique comparison between survivors’ perceptions of exercise programming (preferences, barriers and facilitators), prior to taking part in a formal community-based exercise program, to perceptions of survivors after having participated in the program. It is interesting to note that pre-ACE SEG participants reported concerns of increased risk of injury and lack of familiarity with exercise equipment as barriers to exercise, whereas post-ACE, SEG participants did not. In fact, post-ACE participants identified facilitators of ACE as a program that (1) was safe, (2) offered variability in exercise programming and (3) was led by an exercise specialist knowledgeable in cancer. Consistent with the literature, the main barrier of time, along with disease and treatment symptoms, and accessibility were reported by SEG participants at both pre- and post-ACE SEG time points [28,29].

Post-ACE benefits included maintenance or reduction in symptoms as shown in the post-study questionnaire findings, and motivation stemming from (1) survivor-peers, (2) an onsite knowledgeable exercise specialist and a physical therapist, and (3) baseline and post-study fitness assessments. Access to a physical therapist was found beneficial in tailoring exercise to address cancer therapeutic benefits (e.g., managing cancer-related fatigue, lymphedema, chemotherapy induced peripheral neuropathy) and other existing musculoskeletal concerns [30]. There is a growing discussion in the literature regarding a lack of education connecting exercise benefits specifically to symptom management and survival benefits, with survivors receiving only general education, in turn leading to poor exercise adherence partly from not fully understanding the benefits [11].

Recent evidence, however, suggests that increased effort be placed on addressing barriers rather than emphasizing exercise benefits [31]. From the survivor perspective, the complexity of barriers to exercise may outweigh any potential benefits. New barriers identified by this study included concerns over exposure to germs in a public facility (for those on active treatment), and accessibility issues due to the downtown program location, and associated parking costs. Thus, to improve exercise uptake, consideration should be given to address specific barriers relevant to the survivor and their phase in the cancer care continuum (pre-treatment, treatment, post-treatment, survivorship, remission, palliative), and efforts are needed to enhance exercise counselling practice and uptake [32].

Survivor exercise preferences have been discussed in the literature as wide ranging and diverse [33]. Commonly cited preferences in the literature of moderate-intensity exercise and flexible programming (e.g., including home-based options) were also reflected in our post-ACE questionnaire, with 94% preferring moderate-intensity exercise and 82% preferring a combination of supervised and unsupervised exercise [33].

Of note is research emerging from the perspective of the HCP toward exercise counselling and referral for cancer survivors. The diagnosis of cancer itself represents a ‘teachable moment’ to introduce health behavior change [34]. However, HCP exercise recommendations have been found to be general, rare and inconsistent, despite survivors expressing the need for more exercise counseling in the clinical setting [35]. A survey of 120 Canadian oncologists found that 80% were unaware of any exercise guidelines for cancer survivors, and had a lack of knowledge on screening and identification of appropriate survivors for exercise referral [36]. HCPs also cite a primary barrier of time in clinic, along with a lack of role definition in the responsibility for exercise education, uncertainty on optimal timing to initiate such a discussion, and perceiving a negative attitude towards exercise from survivors [35,37]. In combination with our findings, a gap in exercise counseling of cancer survivors appears to exist both from the perspective of the survivors and HCPs. It is important to note the significant influence that HCPs, especially oncologists, can have on patients’ exercise participation by giving effective, timely exercise advice and education [26,38]. Further work is needed to bridge the exercise knowledge gap for both survivors and HCPs.

### Limitations

A primary limitation of the present iKT study was the small convenience sample, with results limited in generalizability to primarily the breast cancer tumor group and the single setting; however, we reached data saturation, suggesting that our sample was adequate for our knowledge translation purposes. While there was less representation from other tumor groups, this was not surprising given that 50% of ACE trial participants (*n* = 40) were breast cancer survivors, and evidence related to exercise to date is largely based on data collected in the breast cancer population [4]. The post-ACE questionnaire was optional for participants, limiting quantitative findings and overall generalizability. Caregivers were also not included in the post-ACE SEG sessions, limiting our findings to participants alone. For future research, consideration should be given to sampling of other common tumor types such as prostate, colon and lung. The SEG sessions were only held at the Edmonton location, as similar work with breast and prostate cancer groups had been completed or was in process at the Calgary site [39,40]. A further limitation was that the SEG sessions were not audio recorded. Audio recordings would have allowed us to ensure that important details were not missed; however, sessions included written feedback from survivors and independent observers were used to record the small group discussions. The post-ACE satisfaction questionnaire was anonymous; thus, we were limited in our ability to connect findings to participant characteristics of tumor type, gender, and age. This anonymity was to ensure that participants were comfortable in sharing their true experience and providing meaningful feedback.

## 5. Conclusions

Utilizing an integrated knowledge translation approach helped us to inform acceptable ACE program implementation and optimize survivor program satisfaction. A supported exercise program involving both a cancer-specific trained exercise specialist and physiotherapist may prove beneficial for addressing both physical fitness and cancer-related impairments simultaneously. The findings of this study were used to inform the ACE Hybrid Effectiveness-Implementation Study, with key actionable initiatives such as ensuring easily accessible community locations. To address concerns with exposure to bacteria and viruses in a public facility, potential actionable options could include supported home-based exercise, flexible programming options and/or entry into community-based exercise programs in the post-treatment time period. Given the patient-identified gap in communication about exercise for survivors, further research exploring the perspective of HCPs on exercise counseling and referral practices is likely the next critical step to inform integration of exercise into cancer care.

## Figures and Tables

**Figure 1 ijerph-17-03911-f001:**
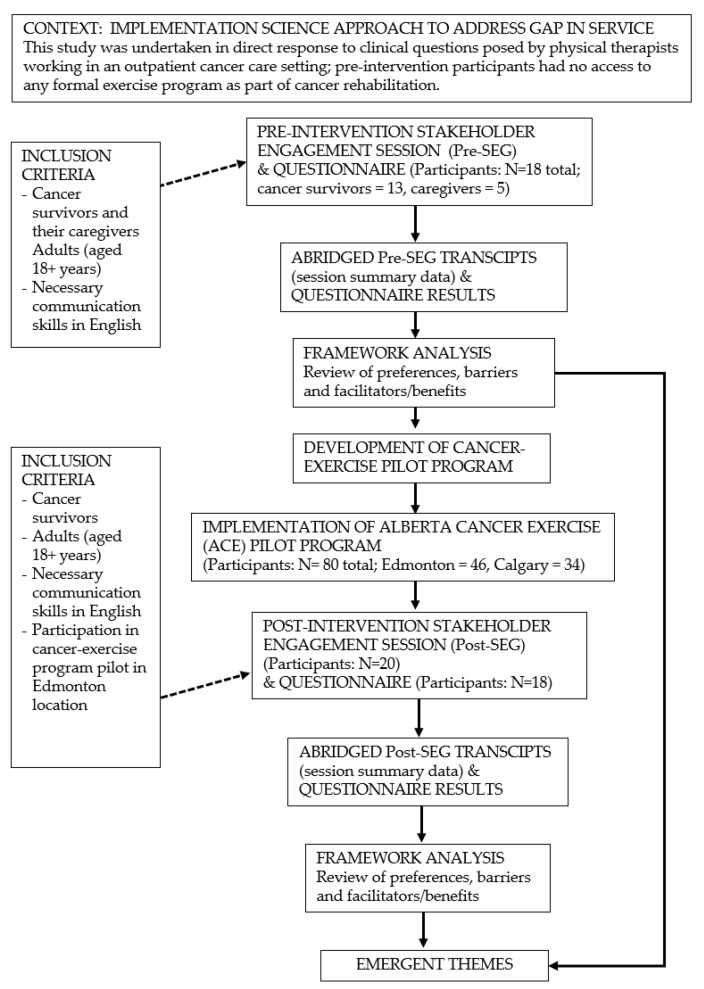
Study Schema.

**Figure 2 ijerph-17-03911-f002:**
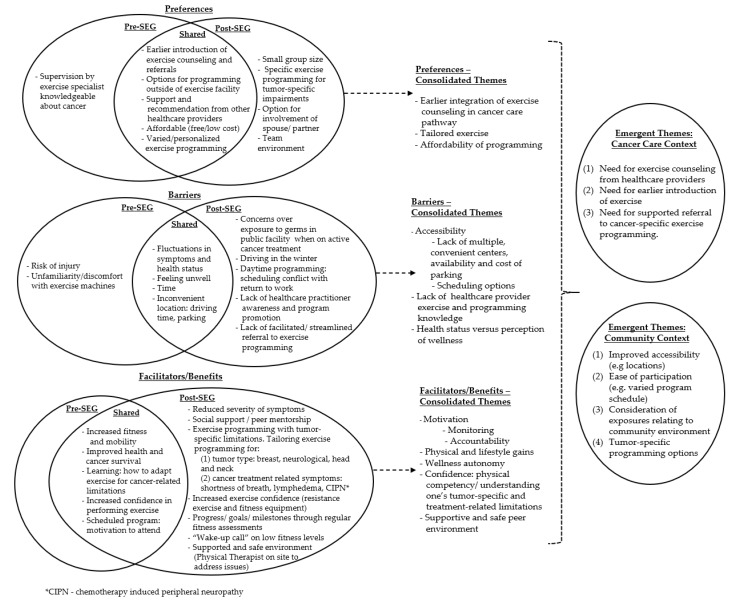
Framework analyses of preferences, barriers and facilitators/benefits pre- and post-stakeholder engagement groups to reach common themes.

**Table 1 ijerph-17-03911-t001:** Baseline demographic and medical data.

Participant Characteristics	Pre-Intervention	Post-Intervention
	(*n* = 13; *n* = 5), No. (%)	(*n* = 20), No. (%)
**Sex**		
Male survivor	4 (31)	3 (15)
Caregiver	1	
Female survivor	9 (69)	17 (85)
Caregiver	4	
**Age**		
26–39	2 (15)	1 (5)
40–54	4 (31)	6 (30)
55–69	6 (46)	7 (35)
>70	1 (8)	6 (30)
**Tumor Type**		
Breast	7 (54)	12 (60)
Caregiver	1	
Head and neck	3 (23)	4 (20)
Caregiver	1	
Lymphoma	1 (8)	1 (5)
Gastrointestinal	2 (15)	2 (10)
Caregiver	1	
Prostate	-	1 (5)
Other	1	-
Caregiver multiple myeloma		
**Cancer Treatment**		
Surgery, radiation, chemotherapy	7 (54)	9 (45)
Surgery, radiation	2 (15)	7 (35)
Surgery, chemotherapy	2 (15)	2 (10)
Surgery alone	1(8)	1(5)
Chemotherapy alone	1(8)	1(5)
**Location of Residence**		
Edmonton (urban)	12 (67)	16 (80)
Within 100 km of Edmonton	5 (28)	4 (20)
Rural > 100 kms	1(5)	-

**Table 2 ijerph-17-03911-t002:** Pre-ACE exercise preferences, barriers and facilitators/benefits.

**Preferences Related to Exercise Programming: *n* = 13 Survivors and *n* = 5 Caregivers**			
Where would you prefer to exercise?	Rank: 1. Community based, 2. Home based		
What type of exercise would you like to do?	Rank: 1. Aerobic, 2. Walking, 3. Resistance exercise		
How many times a week would you like to exercise?	Rank: 1. Two times per week; 2. Three times per week		
How long would you like each session to last?	Rank: 1. One hour per session		
What intensity of exercise would you prefer?	Rank: 1. Mild to moderate, 2. Moderate		
Who would you prefer to exercise with?	Rank: 1. Other cancer survivors, 2. Partner/support person		
**Preferences Related to Exercise Counseling: *n* = 13 (%)**			
Did you receive counseling about exercise at any point from diagnosis to treatment completion?	Not discussed: 10 (77)	Survivor-initiated discussion: 0 (0)	Oncologist-initiated discussion: 3 (23)
Would you have preferred to be counseled about exercise?	Yes: 11 (85)	Maybe: 0 (0)	No: 2 (15)
When would you prefer this counseling?	Before/during treatment: 2 (15)	After treatment: 2 (15)	Multiple time points: 9 (69)
Where would you prefer this counseling to happen?	Cancer centre: 7 (54)	Community fitness centre: 2 (15)	Any location: 4 (31)
Who should provide the counseling?	Exercise specialist: 8 (62)	Health care provider: 3 (23)	Other cancer survivor: 2 (15)
What would be your preferred format of counseling?	Face to face: 10 (77)	Written materials: 2 (15)	Other: telephone/internet: 1 (1)
**Barriers to Exercise (*n* = 18)**	**Facilitators (*n* = 18)**	**Benefits (*n* = 18)**	
Lack of time Instructors unfamiliar with cancer Injury risk Intimidation/insecurity in public facility Feeling unwell Symptoms, e.g., fatigue Financial: cost of program Lack of personalized exercise programming	Instructor who is knowledgeable Instructor who is fun Convenient location Free parking, public transit access Support of other health care providers: nurse, physical therapist	Increases fitness Better health and overall survival Improved mobility Emotional support Better mental health Confidence in exercise	

**Table 3 ijerph-17-03911-t003:** Post-ACE preferences, barriers, facilitators/benefits and satisfaction questionnaire.

**Top Preferences Related to Exercise Programming (*n* = 18; Counts When More Than One Option Allowed)**			
Location of program (*n* = 21)	Community fitness centre: 12 (57%); home/outdoors: 7 (33%)		
Type of exercise (*n* = 22)	Combined aerobic and resistance: 18 (82%)		
Frequency and Duration (*n* = 22)	2× or 3× per week: 16 (72%)		
Duration (*n* = 22)	50–60 min sessions: 16 (72%)		
Intensity (*n* = 22)	Moderate intensity (not exhausting, light perspiration): 17 (77%)		
Who would you prefer to exercise with? (*n* = 24)	Other cancer patients: 13 (54%), spouse/friend: 9 (38%)		
**Barriers to Exercise Participation (*n* = 18)**			
	Never	Rarely–Occasionally	Often–Very Often
Fatigue	4 (22)	11 (61)	3 (17)
Lack of enjoyment	11 (61)	5 (28)	2 (11)
Lack of self-discipline	6 (33)	11 (61)	1 (6)
Pain	7 (39)	9 (50)	2 (11)
Weather	12 (67)	4 (22)	2 (11)
Exercise is boring	14 (78)	4 (22)	-
Muscle weakness	6 (33)	7 (39)	5 (28)
Lack of time	12 (67)	5 (28)	1 (6)
Lack of confidence in exercise abilities	14 (78)	4 (22)	-
**Facilitators/Benefits of Exercise program on (*n* = 17)**			
	Very Much—Slightly Worse	No Change—Slightly Improved	Somewhat—Very Much Approved
Physical functioning	-	2 (12)	15 (88)
Muscle strength	-	3 (18)	14 (82)
Overall quality of life	-	4 (24)	13 (76)
Fatigue	-	6 (35)	11 (65)
Energy level	-	6 (35)	11 (65)
Ability to perform ADL	-	7 (41)	10 (59)
Treatment recovery	-	8 (47)	9 (53)
**ACE Exercise Program Satisfaction (*n* = 17)**			
	Not at All	A Little Bit/Somewhat	Quite a Bit/Very Much
How beneficial was the program?	-	-	17 (100)
How enjoyable was the program?	-	2 (12)	15 (88)
How supportive were your friends and family?	-	2 (12)	15 (88)
How motivated were you (to?) participate?	-	3 (18)	14 (82)
How difficult was it for you to participate?	6 (35)	11 (65)	-
How difficult will it be for you to continue to exercise?	9 (53)	7 (41)	1 (6)

**Table 4 ijerph-17-03911-t004:** Translation of themes to actionable items.

CONTEXT	THEME	ACTIONABLE ITEMS
Cancer Care Setting	Need for exercise counseling from health care provider	- Determine health care provider preferences/barriers/facilitators towards exercise counseling and referral of cancer survivors to community-based exercise. - Incorporation of exercise counseling and referral by health care providers into cancer care pathway - Facilitate referral pathway from cancer care setting to community facility (e.g., safety screening, medical approval as needed, consent for sharing of personal information)
	Need for earlier introduction of exercise	
	Need for supported referral to cancer-specific exercise programming	
Community-Based Setting	Improved accessibility (e.g., locations)	- Consideration of accessible locations across provincial regions: expand community fitness partnerships - Development of ongoing relationship with community facilities
	Ease of participation (e.g., varied program schedule)	Where possible offer varied scheduling of cancer-specific exercise programming (e.g., morning/afternoon/evening options)
	Consideration of exposures relating to community environment	- Consideration of non-public facilities, home-based and/or flexible programming - Consideration of entry to community programming post-cancer treatment
	Tumor-specific programming options	Development of subset programming to meet specific needs (e.g., breast, head and neck, neurological tumor-specific exercise programming)

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
