# Peer review of "A Practical Approach to Using Integrated Knowledge Translation to Inform a Community-Based Exercise Study"

_ijerph, 2020, doi:10.3390/ijerph17113911_

Round 1

Reviewer 1 Report

Please see attached article review.

Reviewer 2 Report

This project was developed to understand cancer survivors needs prior to and following an exercise intervention. The goal is to evaluate the program in preparation of moving the intervention to a province-wide program in Canada. This is a well-defined project with clear aims. The project is important in terms of better understanding how to translate research findings related to the importance of exercise for cancer survivors to an actual community based setting. Focusing on knowledge translation is critically important at this point in our research process. Using person reported outcomes is required to understand the acceptance or non-acceptance of an evidence based intervention in a “real life” setting. There are some significant weaknesses in the paper. The primary weakness is a lack of detail in the methods (particularly related to number and selection of participants). There is an additional weakness in the findings where a table (#4) provides important  information about findings that are not shared in the paper (i.e. focus on specific types of tumors for certain actionable steps).

TITLE                                              

- The title clearly conveys the point of the article.

ABSTRACT

The abstract is accurate, concise, and clearly describes the study.

INTRODUCTION

This is an appropriate introduction to the overarching study and the KT plan and importance.

Purpose of the project is clearly stated

METHODS          

- The Study design was appropriate to achieve the project objective. The methods are clearly described, and the process of the project outlined.  Person reported outcomes/ experiences are necessary to collect in order to understand what worked and what did not work in terms of the intervention. That information will guide any modification to be made before moving the intervention into the community for broad translation.

The Study population was clearly and adequately described.

The qualitative analysis methods were used appropriately although there is no discussion related to the number of participants (see comment below).

The quantitative component of this project is not well described. There is questionnaire data collected but apparently only from 18 participants making any quantitative findings probably lacking but this data does only seem to be used descriptively which can be appropriate. However, this needs to be better described and explained. I do not see any mention of the questionnaire in the data collection section of the paper. It is mentioned in the data analysis section though.

There is no explanation for the number of participants included in this project. Beyond being a quantitative weakness, qualitatively there should be some reason why the project was completed with 38 participants (combing the pre and post data collection times). Additionally, not all involved in the intervention completed the post questionnaires. Was there a reason for the numbers? How were they selected from the intervention group for the post data collection? Was saturation met? Is there a reason why no caregiver feedback was collected in the post data collection phase?

There needs to be more detail in the methods section.

RESULTS                                                              

- The results are clearly presented. There is an appropriate use of quotes and the Figure 2 is very helpful and interesting. The information in Table 4 is very helpful as well. It is nice to have the findings connected to action.

In Table 4 there is a bullet for incorporating exercise counseling and referral into tumor-specific care pathways. Was this information found in this project? I do not see and connecting of findings to certain types of cancer. This is unclear to me but would be very important information to add. There is a second comment about subsets of programs to meet specific needs and that is not shown in the finds data either.

DISCUSSION

This was appropriate

- Limitations of the study were noted, but in them, it is stated this was a “purposively” selected sample size. This is the information that is missing in the methods. How was this purposively selected, how was the sample size decided upon? These is very important information in terms of rigor.

CONCLUSIONS

-The conclusion was clearly stated and concise.  

FORM, STYLE, AND SUBSTANCE

This is a well-organized and cohesive paper.

Round 2

Reviewer 2 Report

Thank you for addressing the original comments. I think your edits have taken care of any concerns.